# Influence of Sodium Hypochlorite Treatment on Pore Size Distribution of Polysulfone/Polyvinylpyrrolidone Membranes

**DOI:** 10.3390/membranes10110356

**Published:** 2020-11-19

**Authors:** George Dibrov, George Kagramanov, Vladislav Sudin, Evgenia Grushevenko, Alexey Yushkin, Alexey Volkov

**Affiliations:** 1D.Mendeleev University of Chemical Technology of Russia, 125047 Moscow, Russia; kadri@muctr.ru; 2Baikov Institute of Metallurgy and Materials Science, Russian Academy of Sciences, 119334 Moscow, Russia; sudin.vlad@gmail.com; 3A.V. Topchiev Institute of Petrochemical Synthesis RAS, Russian Academy of Sciences, 119991 Moscow, Russia; evgrushevenko@ips.ac.ru (E.G.); halex@ips.ac.ru (A.Y.); avolkov@ips.ac.ru (A.V.)

**Keywords:** polysulfone, polyvinylpyrrolidone, membrane, ultrafiltration, porosimetry, sodium hypochlorite

## Abstract

This work was focused on the study of hypochlorite treatment on the pore size distribution of membranes. To this end, ultrafiltration membranes from a polysulfone/polyvinylpyrrolidone blend with a sponge-like structure were fabricated and exposed to hypochlorite solutions with different active chlorine concentrations for 4 h at ambient temperature. Liquid–liquid displacement and scanning electron microscopy were employed to study the limiting and surface pores, respectively. After treatment with 50 ppm hypochlorite solution at pH = 7.2, a five-fold increase in water permeance up to 1400 L/(m^2^·h·bar) was observed, accompanied by a 40% increase in the limiting pore sizes and almost a three-fold increase in the porosity. After 5000 ppm treatment at pH = 11.5, a 40% rise in the maximum limiting pore size and almost a two-fold increase in the porosity and permeance was observed, whereas the mean pore size was constant. Apparently, changes in the membrane structure at pH = 11.5 were connected with polyvinylpyrrolidone (PVP) degradation and wash-out, whereas at lower pH and despite lower active chlorine concentration, this process was coupled with polysulfone (PSf) destruction and removal.

## 1. Introduction

Polysulfone (PSf) is a widely applied material to produce polymeric membranes for different applications [1,2,3,4,5,6,7,8,9,10,11,12,13,14,15,16,17,18]. In water treatment, these membranes are commonly modified by the addition of polyvinylpyrrolidone (PVP) to increase hydrophilicity and water permeance [9,19,20,21,22,23]. PVP possesses good miscibility with PSf and dope solution components and good solubility with the major coagulant–water, which facilitates modification by a trivial blending of these two polymers in the dope solution. Along with the improvement of membrane transport characteristics, PVP decreases membrane liability to fouling by organic substances [24,25,26,27].

To maintain the required performance during long-term operation, regular chemical washing, and disinfection of the filtration membranes, are required. Sodium hypochlorite (NaOCl) is a commonly used chemical agent due to its low cost and high effectiveness. Depending on the characteristics of the treated water, membrane elements may be subjected to NaOCl solutions with active chlorine concentrations up to 20 ppm for up to 5 min for backwashing processes and with concentrations up to 400 ppm for up to 4 h for chemical cleaning processes [28].

A change in the membrane characteristics after the hypochlorite treatment can occur due to PVP degradation and its leaching from the membrane, as well as due to the partial destruction of the PSf matrix [20]. There are two mechanisms responsible for the PVP degradation: polymer chain scission and opening of the pyrrolidone ring of the PVP [29]. PSf chains are mainly damaged by the HClO form of chlorine, which mainly exists at pH 7–10 [20].

The treatment of membranes made of a PSf/PVP blend with hypochlorite solution may increase permeance and reduce retention [20,30,31,32,33]. Exposure to a hypochlorite solution with active chlorine concentration of 4000–6000 ppm at pH = 11.5 for 48 h elevated PSf/PVP (15/5 wt.%) membrane water permeance by 4–5.4 times up to 760 L/(m^2^·h·bar) [31]. At the same time, PVP (10–40 kDa) retention declined, while bovine serum albumin (BSA, 67 kDa) retention was constant. In the work [32], dialysis membranes of PSf were treated with hypochlorite solution with an active chlorine concentration of 4% for 1 h. Based on the dextrans retention data, it was calculated that such treatment increased the effective pore size from 0.76 to 1.58 nm. Similar tendencies were observed for other polymeric membranes containing PVP as an additive, namely for polyethersulfone [28,34,35,36,37], polyvinylidene fluoride [38,39], cellulose acetate [40] and polyacrylonitrile [41]. Thus, it should be taken into consideration that exposure to sodium hypochlorite solution can modify PSf/PVP membrane properties.

In the work [34], pore size distribution was calculated using data on the retention of dextrans for polyethersulfone/PVP membranes treated with 350 ppm active chlorine at pH = 8. In this work, pore size distribution of PSf/PVP membranes before and after hypochlorite treatment was determined directly, employing liquid–liquid displacement porosimetry and scanning electron microscopy. These techniques allow for the measurement of limiting and surface pores, respectively. The evolution of membrane properties was correlated with the active chlorine concentration and pH of the solution. The aim of this study was the evaluation of the effect of sodium hypochlorite treatment on the porous structure and transport characteristics of ultrafiltration membranes made of a polysulfone/polyvinylpyrrolidone blend.

## 2. Materials and Methods

### 2.1. Materials

Polysulfone Ultrason S6010 natural (M_w_ = 50 kDa) and polyvinylpyrrolidones Luvitec K30 (M_w_ = 50 kDa) and Luvitec K90 (M_w_ = 1400 kDa) were provided by BASF (Ludwigshafen, Germany). The following reagents were used in this work: polyethylene glycol 400 (PEG-400, TC 2226-061-05766801-2006, Forvard Group, Moscow, Russia), sodium hypochlorite (grade “A”, 170–190 g/L, Russian Chemist, Moscow, Russia), ethanol (EtOH, 95%, Spirtmed, Moscow, Russia). Reverse osmosis water with a specific conductivity of no more than 30 μS/cm was produced on-site. The following extra pure (>99.5%) reagents, provided by CHIMMED, Russia were employed: *N*,*N*-dimethylacetamide (DMAc), isobutanol (BuOH), glycerin, sodium bisulfate.

### 2.2. Membrane Spinning and Characterization

Polysulfone hollow fibers were obtained by the non-solvent-induced phase separation method [42]. The dope solution consisted of the polymer, DMAc was selected as a good solvent and PEG-400 as a non-solvent due to good pore-forming properties [43]. PVP with different molecular weights was employed to increase hydrophilicity, while PVP K-30 was used as a pore-forming agent and PVP K-90 was used to increase the solution viscosity [21].

The polymer content was chosen to be 20 wt.% to maintain good mechanical properties of the membrane. The composition of the dope solution was selected in accordance with two criteria. Firstly, the homogeneous dope solution should be as close as possible to the “binodal” boundary, which separates single-phase and two-phase regions on the three-component phase diagram. Secondly, the dope solution should have a viscosity between 30 and 100 Pa·s at 25 °C. An insufficient value of viscosity may induce the formation of finger-like “macrovoids” which can lead to the appearance of defects in the fiber. On the other hand, when the viscosity value exceeds 100 Pa·s, the dope solution is hard to prepare, degas and spin into fiber [7]. Therefore, the resulting solution contained 57 wt.% of DMAc, 19 wt.% of PEG-400, 2 wt.% of PVP K-30 and 2 wt.% of PVP K-90.

The dope solution was prepared in a round-bottom flask, placed in a silicon oil bath. The flask was equipped with a dynamic double seal stirrer bearing to prevent exchange with the atmosphere during mixing. The solvent was placed in the flask and heated up to 90 °C under stirring for 20 min. Over the next two hours, the polymer was slowly added to prevent granule coagulation and caking. Afterward, PVP and PEG-400 were added and the solution was stirred for two hours. The sample was taken from the solution to measure the viscosity at 25 °C using a falling-sphere viscosimeter. The dope solution was transferred into the tank, equipped with a water jacket, vacuumed for one minute at 0.1 bar and left under vacuum for 16–20 h for degassing at 50 °C.

The schematic of the setup is described in detail in [44]. The dope solution temperature was kept at 25 °C, employing the water jacket. The dope solution was supplied to the spinneret using the gear pump. The schematic of the spinneret is presented in the work [7]. The ring duct diameter of the spinneret was 1.5 mm and the outer bore needle diameter was 1 mm. A water solution of DMAc (50 wt. %) at room temperature was employed as a bore liquid. The height of the air gap was 0.6 m, the coagulation bath was filled with water at room temperature and the take-up speed was 7 m/min. The selective layer of these membranes was situated at the lumen side of the fiber.

Fiber bundles were cut and placed in a rinsing bath with running water for at least 24 h. Afterward, fibers were placed and stored in a preservative water solution, containing 50 wt.% of glycerin and 1 wt.% of sodium metabisulfite. As required, fibers were thoroughly rinsed with water and used. Whenever it was necessary to remove water from the fibers, they were placed in ethanol for 2 h and dried in air for at least 4 h.

Aqueous solutions of sodium hypochlorite were prepared gravimetrically and the pH value was determined using an HI 98,127 pH-meter (HANNA Instruments, Woonsocket, RI, USA). Washed fibers were placed in the hypochlorite solution for 4 h at room temperature and then rinsed with water.

A Levenhuk 40L NG microscope equipped with a C310 NG 3M digital camera was employed to determine the lumen and outer diameters of the fiber. To obtain a smooth cross-section of the fiber, freeze-fracturing in liquid nitrogen was performed after conditioning in ethanol for half of an hour.

A high-resolution scanning electron microscope Supra 50 VP LEO (Carl Zeiss SMT Ltd., Oberkochen, Germany) was used at an accelerating voltage of 5 kV and an aperture of 30 μm to study the microstructure of asymmetric hollow fibers. Samples were sputtered with a 5–15 nm layer of gold to withdraw the excess charge. Gwiddion software was employed to examine the microphotographs.

The bubble point technique was used to characterize the mechanical properties, fiber integrity, and the absence of defects. To this effect, fibers were submerged in the water bath and their lumen side was pressurized with nitrogen up to 10 bar. The visible absence of stable gas bubble flows indicated that the membrane possessed neither defects nor large pores.

### 2.3. Measurement of Pore Size and Distribution

The pore size distribution (PSD) was measured by liquid–liquid displacement porosimetry (LLDP) [45] using a POROLIQ 1000 ML porometer (Porometer, Nazareth, Belgium). The measurements were carried out at 25 °C by using a pair of immiscible liquids prepared by the demixing of isobutanol and water blend (1/4, *v*/*v*). The alcohol-rich phase was used as the wetting liquid and the water-rich phase was used as the displacing liquid. Samples were placed into the beaker with the wetting liquid for at least 2 h at 20 °C before the testing. The operating principle is based on the measurement of the equilibrium pressure corresponding to the flux of the displacing liquid. The displacement of the wetting liquid was carried out by a stepwise increase in the trans-membrane pressure with monitoring of the flux through the membrane after a 180 s initial stabilization time at each applied pressure. The attainment of linear dependence of the flux from pressure indicated a complete displacement of the wetting liquid. The following sharp increase in the linear relation angle denoted the bursting of the fiber. The diameter (*D*) of the open pore is related to the trans-membrane pressure via the Young–Laplace equation:(1)$$D=\frac{4\cdot \gamma \cdot \cos\theta }{\Delta p}$$
where *γ* is the interfacial tension between the two liquids, *θ* is the contact angle between the membrane and the wetting liquid, Δ*p* is the trans-membrane pressure. Interfacial tension *γ* for the mixture of isobutanol and water is 1.9 mN/m at 25 °C.

To determine the wetting contact angle, flat asymmetric membranes were obtained by casting the dope solution on a smooth glass surface, distributing it with a casting knife and consequent coagulation in water. The same dope solution and post-treatment were employed for flat membranes to reproduce the amount and composition of residual solvents, and morphology at the inner surface of the fiber. Flat membranes were rinsed for at least 24 h with running water. The wetting contact angle was measured by employing Easy Drop DSA 20 equipment. Surface pores and roughness can increase the rate of contact angle advance. To this end, the value was recorded 2 s after the placement of the drop.

To assess the pore size distribution (PSD), the ratio of the two-liquid flux (i.e., the flux of the displacing liquid in the presence of the wetting liquid) to the one-fluid flux (the flux of the displacing liquid only) is calculated:(2)$$F(\Delta p)=\frac{J_{2}(\Delta p)}{J_{1}(\Delta p)}$$

Function F(∆*p*) can be converted to the function F(*D*) using Equation (1). The function F(*D*) represents the fraction of open pores with a diameter greater than or equal to *D*. In other words, it is nothing more than a cumulative distribution function. The differential function of the distribution of flow-through pores, f(*D*), i.e., the PSD, equals the (negative) derivative of F(*D*):(3)$$f(D)=-\frac{dF(D)}{dD}$$

The average pore diameter was calculated:(4)$$\langle D\rangle =\int_{0}^{\infty }D\cdot f(D)dD$$

## 3. Results

The resulting fiber cross-section micrograph is presented in Figure 1. The outer diameter was 1.24 mm, the inner diameter was 0.84 and the wall thickness was 0.2 mm. The fiber possessed a sponge-like structure, which is beneficial to good mechanical properties of the fiber due to the absence of finger-like pores. Such finger-like pores, also called “macrovoids”, are often responsible for the appearance of defects, especially after long-term operation of fibers in the filtration–backwash mode.

The dependence of membrane permeance from the active chlorine concentration in the hypochlorite solution and pH is presented in Table 1. Pure water permeance of the initial fiber was 270 L/(m^2^·h·bar). Similar to works [20,28,33], the treatment of the membranes with the hypochlorite solution led to a rise in permanence. At the active chlorine concentration of 50 ppm, permeance increased up to 1400 L/(m^2^·h·bar) (Table 1), which is the maximum value of permeance attained after hypochlorite treatment. For higher active chlorine concentrations, permeance values gradually decreased but remained higher than the initial membrane permeance.

Using the LLDP method, limiting pore size distributions were determined for initial and hypochlorite-treated membranes (Figure 2). After the treatment of membranes with a 50 ppm hypochlorite solution, both the maximum and mean pore size rose significantly, whereas, after 5000 ppm, the maximum pore size increased, but the mean pore size showed virtually no change. The membrane porosity (Table 2) was calculated using the Hagen–Poiseuille equation, values of pure water permeances and mean pore diameters. Pores were assumed to have a straight cylindrical shape: pore length was equal to the fiber wall thickness and the value of tortuosity was taken as 1.

SEM micrographs of the lumen and shell sides of initial and hypochlorite-treated membranes are presented in Figure 3. Employing the grain analysis tool of Gwiddion porosity, maximum and mean pore sizes were estimated (Table 2). The results obtained employing LLDP and SEM techniques are in good agreement. The fact that the mean limiting pore size is close to the mean inner pore size and less than half of the mean outer pore size proves that the selective layer is situated close to the lumen side of the fiber.

## 4. Discussion

The obtained results revealed an obvious influence of hypochlorite treatment on pore sizes, permeance and water contact angle of membranes in the PSf/PVP blend.

In the works [33,46], hypochlorite treatment led to a decrease in the water contact angle. Such surface hydrophilization was ascribed to partial ionization on the membrane surface and enlargement of the surface pores. In this work, the water wetting contact angle of the untreated membrane was 50° due to the addition of PVP to the dope solution. Besides, surface pores and roughness can decrease the value of the contact angle if the material is hydrophilic. After treatment with 5000 ppm hypochlorite solution, the contact angle rose (Table 2) up to 90°, close to the water wetting contact angle for pure PSf [30,47]. The same tendency was also observed in work [30] and can be connected with the partial destruction and elimination of PVP from the membrane surface.

Typically, an ultrafiltration hollow fiber membrane should withstand pressure of 1.5 bar during filtration and 3 bar during backwashing. In this work, owing to the sponge-like microstructure, the burst pressure of the initial membrane in the water exceeded 10 bar (Table 2).

Plasticization by isobutanol reduced the mechanical properties of the hollow fiber, which made the burst pressure in the butanol–water solution lower than in water. Treatment with a 50 ppm solution at pH = 7.2 resulted in a two-fold decrease in the burst pressure (Table 2). Such loss of mechanical properties can be connected with the molecular degradation of PSf chains. In the work [20], a drift to lower PSf molecular weight was also observed at pH 7–10, which occurred due to the presence of the HClO form of the chlorine [20]. The fiber burst pressure in water still exceeded 10 bar, which proves that after such an exposition mode, these fibers are still applicable for water treatment.

The value of burst pressure remained nearly the same after the treatment of the membrane with the 5000 ppm solution at pH = 11.5. Despite high active chlorine concentration, PSf degradation did not take place, because at a pH higher than 11, the HClO form of chlorine is absent in the solution.

Changes in the pore sizes and porosities were less dramatic at the lumen surface of the fiber due to the static conditions of the experiment. For instance, for 5000 ppm, a 2.5-fold increase of the outer surface porosity was observed, while the inner surface porosity rose only by 40%, and a two-fold increase of the limiting porosity was detected (Table 2). During 4 h of the exposition without mixing, the outer surface was more easily accessed by the hypochlorite solution.

A 40% increase in the limiting pore sizes and almost a three-fold increase in the porosity were observed after 50 ppm exposition (Table 2). Such behavior can be associated with the simultaneous destruction and wash-out of PSf and PVP from the membrane matrix at the contact interface with the chlorine solution. This hypothesis is confirmed by the fact that despite such a large increase in the porosity, the water wetting contact angle grew only from 50° to 60°.

After the 5000 ppm treatment, a 40% rise of the maximum limiting pore size and almost a two-fold increase in the porosity was observed, whereas the mean pore size was constant (Table 2). Such behavior is connected with PVP leaching from the membrane surface, which is confirmed by the elevation of the water wetting angle from 50° to 90°. No significant change in the burst pressure confirms that PSf matrix degradation did not occur.

The rise of porosity after the 5000 ppm treatment is mostly connected with an increase in the pore size accompanied by the formation of a new group of pores with a smaller diameter. However, after the 50 ppm treatment, the rise in the pore size is caused not only by the growth of the initial pores but also by fusion of some pores into large ones (Figure 2 and Figure 3).

The increase in pore sizes and porosity resulted in the rise of water permeance with the maximum observed after the 50 ppm hypochlorite treatment. The rise was not so dramatic after the 5000 ppm treatment not only because the mean pore size was preserved, but also because the membrane became less hydrophilic.

The results obtained in this work employing the SEM technique conform to the previously reported data. In the work [40], after the treatment of a cellulose acetate/PVP membrane with 200 ppm hypochlorite at pH = 7 for over 24 h, the surface became smoother and the pore radius increased. After the exposure of the PSf/PVP membrane to a 4000 ppm hypochlorite solution at pH = 7 for over 48 h, the inner surface became rough and the pore size on the outer surface increased [31]. In the work [35], the employment of 5000 ppm NaOCl solution for 100 h at pH = 9 led to a significant increase in the polyethersulfone/PVP membrane pore size. The authors of work [20] noticed that after the treatment of PSf/PVP membranes with 100 ppm of HClO for 16 days at 25 °C, the microscopic morphology changed from filamentous to spongy. In this work, the initial structure of the membrane was spongy and only treatment with 50 ppm led to the transformation of the inner surface from spongy to slightly filamentous due to the simultaneous removal of PSf and PVP. The roughness of the inner and outer surfaces did not vary significantly with the increase in the pore size and porosity.

## 5. Conclusions

In this work, it was shown how the exposure of ultrafiltration membranes PSf/PVP in the hypochlorite solution for 4 h affected the pore size distribution and transport properties of these membranes. Employment of liquid–liquid displacement porosimetry and scanning electron microscopy allowed for simultaneous investigations of limiting and surface pores, respectively.

The conditioning with 50 ppm active chlorine solution resulted in a dramatic increase in permeance, while the burst pressure of the fibers was retained at an acceptable level owing to the sponge-like microstructure. The change of the pore size distribution indicated that membranes after hypochlorite treatment can be still utilized for the retention of colloid particles, large viruses and microorganisms. Therefore, such conditioning in the hypochlorite solution can be employed in ultrafiltration to tailor the membrane properties. Besides, the obtained results should be considered for the conduction of chemical cleaning processes employing sodium hypochlorite.

## Figures and Tables

**Figure 1 membranes-10-00356-f001:**
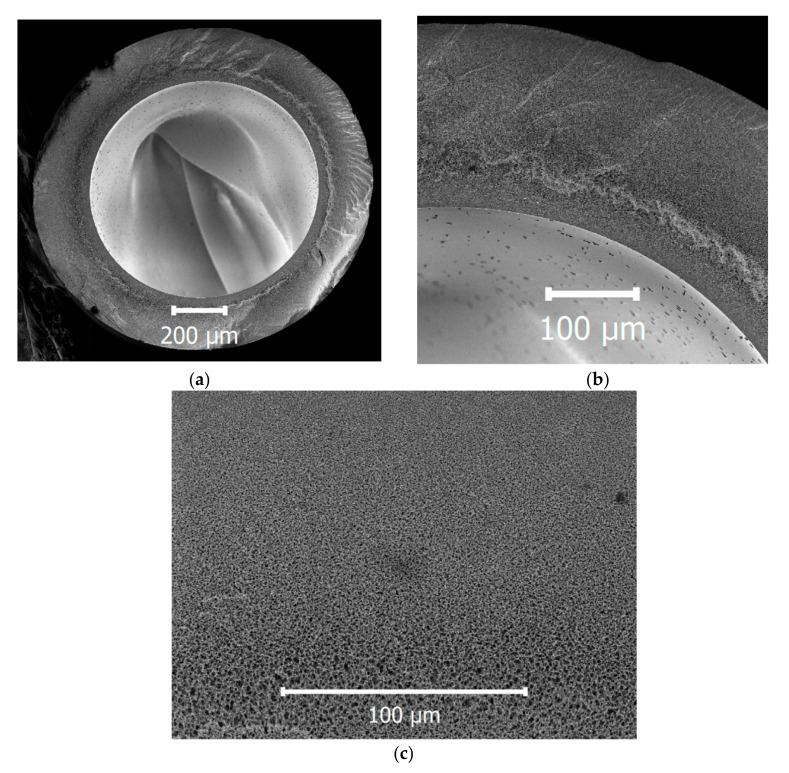
SEM micrographs of the fiber cross-section (**a**) with a zoomed-in view of the wall (**b**,**c**).

**Figure 2 membranes-10-00356-f002:**
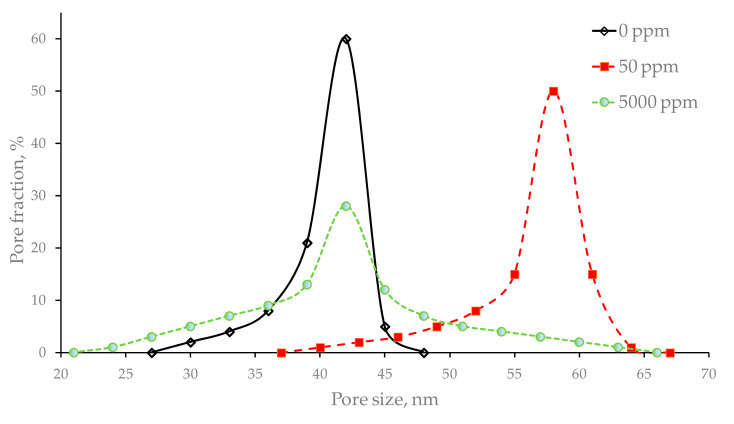
The pore size distribution of the untreated membrane (0 ppm) and membranes treated with solutions with active chlorine concentrations of 50 and 5000 ppm.

**Figure 3 membranes-10-00356-f003:**
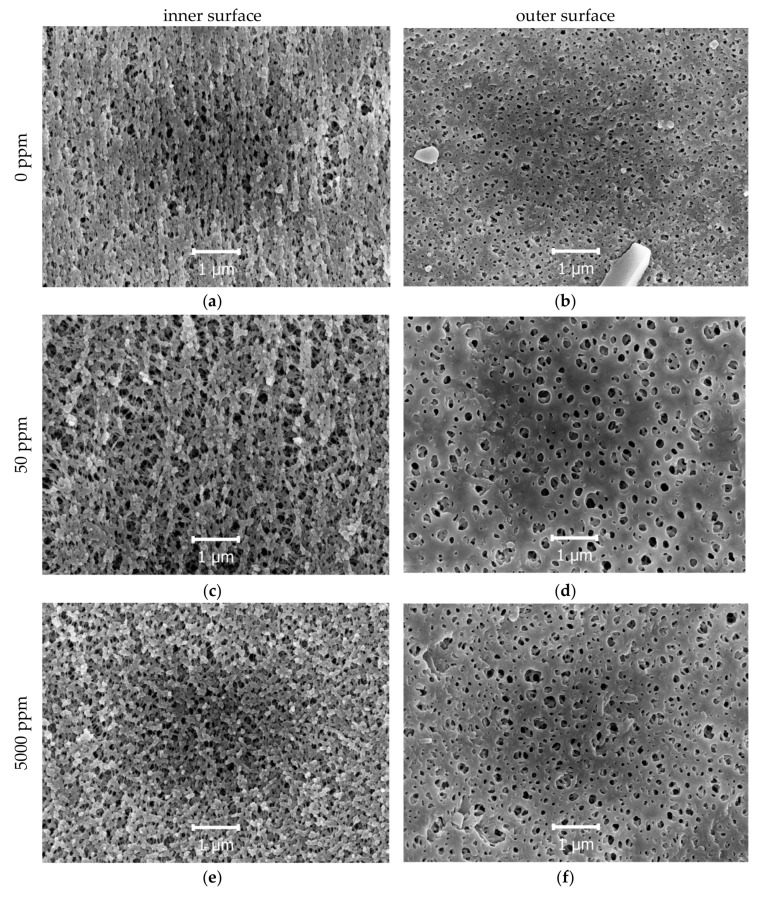
SEM micrographs of the untreated membrane (0 ppm, (**a**)—inner surface, (**b**)—outer surface) and membranes treated with solutions with active chlorine concentrations of 50 ppm ((**c**)—inner surface, (**d**)—outer surface) and 5000 ppm ((**e**)—inner surface, (**f**)—outer surface).

**Table 1 membranes-10-00356-t001:** The dependence of the pure water permeance as a function of the active chlorine concentration (C) and pH value after treatment with the hypochlorite solution. “0” stands for the initial, untreated membrane.

C, ppm	0	25	50	250	500	2500	5000
P/l, L/(m^2^·h·bar)	270 ± 30	1230 ± 90	1400 ± 150	1210 ± 150	810 ± 110	530 ± 70	500 ± 120
pH	6.8	7.0	7.2	9.3	10.2	11.2	11.5

**Table 2 membranes-10-00356-t002:** The dependence of the hollow fiber membrane properties from the active chlorine concentration (C) and pH value after treatment with the hypochlorite solution. “0” stands for the initial, untreated membrane.

C, ppm	0	50	5000
θ(H_2_O), deg.	50 ± 6	60 ± 5	90 ± 5
θ(10% BuOH), deg.	30 ± 1	30 ± 1	30 ± 1
P_burst_(H_2_O), bar	>10	>10	>10
P_burst_ (BuOH/H_2_O), bar	4.17	2.34	4.03
Mean pore size (LLDP), nm	42	58	42
Mean inner pore size (SEM), nm	54	62	56
Mean outer pore size (SEM), nm	93	152	150
Maximum pore size (LLDP), nm	48	67	66
Maximum inner pore size (SEM), nm	120	185	126
Maximum outer pore size (SEM), nm	120	193	190
Porosity (LLDP), %	2.7	7.4	5.1
Inner surface porosity (SEM), %	6.8	12.9	9.3
Outer surface porosity (SEM), %	16.5	43.2	41.9

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
