# Peer review of "Influence of Sodium Hypochlorite Treatment on Pore Size Distribution of Polysulfone/Polyvinylpyrrolidone Membranes"

_membranes, 2020, doi:10.3390/membranes10110356_

Round 1

Reviewer 1 Report

The article of Dibrov et al. discusses an important topic of membrane fabrication, namely
the post-treatment in polysulfone/poly(vinyl pyrrolidone) (PSF/PVP) membranes. The authors investigated
the influence of concentration of active chlorine on the degradation of polysulfone and PVP,
respectively. Both untreated and post-treated membranes were characterized using different techniques.
This study is highly relevant for commercial ultrafiltration membranes. However,
I have some comments which should be clarified first.

1) Figure 1: A SEM micrograph with a higher magnificatio would reveal the sponge-like morphology.

2) Table 1: The unit litre is abbreviated by the symbol "L"

3) More experimental details on the determination of the burst pressure should be given.

4) The authors should comment on the morphology of the flat sheet membranes for measurement of the
contact angle.

5) In Eq. (3), I suggest to write f(D)=-dF(D) / dD

6) For me, Figure 3(a), (c) and (e) look identical, please check.

In particular because of my comment (6), I recommend a major revision.

Author Response

Thank you very much for your time and effort to review our manuscript. We have made our best to improve our manuscript according to your comments. We have expanded Methods, Results and Discussions sections. We have also inserted correct micrographs to Fig. 3. Please note that your comments are in black font and our responses are in red font.

Comment 1.  Figure 1: A SEM micrograph with a higher magnification would reveal the sponge-like morphology.

Answer 1. We have added a micrograph (Fig. 1 c) with higher zoom to the wall, where the sponge-like structure is seen clearer.

Comment 2.  Table 1: The unit litre is abbreviated by the symbol "L"

Answer 2. We have corrected this in the manuscript.

Comment 3. More experimental details on the determination of the burst pressure should be given.

Answer 3. We have extended the experimental section describing the burst pressure test in water (line 122): “To this effect, fibers were submerged in the water bath and their lumen side was pressurized with nitrogen up to 10 bar. The visible absence of stable gas bubble flows indicated that membrane possessed neither defects nor large pores.”

The determination of burst pressure in butanol-water solution is described in section 2.3 (line 136): “The attainment of linear dependence of the flux from pressure indicated a complete displacement of the wetting liquid. The following sharp increase of the linear relation angle denoted the burst of the fiber.”

Comment 4. The authors should comment on the morphology of the flat sheet membranes for the measurement of the contact angle.

Answer 4. We have added comments to the methods section (line 145): “The same dope solution and post-treatment were employed for flat membranes to reproduce the amount and composition of residual solvents, and morphology at the inner surface of the fiber. Flat membranes were rinsed for at least 24 h with running water. The wetting contact angle was measured employing Easy Drop DSA 20 equipment. Surface pores and roughness can increase the rate of contact angle advance. To this end, the value was recorded 2 s after the placement of the drop”

Also, we have added some info regarding this subject to the discussion section (line 202): “In this work, the water wetting contact angle of the untreated membrane was 50° due to the addition of PVP to the dope solution. Besides, surface pores and roughness can decrease the value of the contact angle if the material is hydrophilic.”

Comment 5. In Eq. (3), I suggest to write f(D)=-dF(D) / dD

Answer 5. Corrected as suggested

Comment 6. For me, Figure 3(a), (c) and (e) look identical, please check.

Answer 6. This was a significant mistake from our side, which occurred due to the upload of the wrong version of the manuscript. Thank you for noticing that mistake. We have attached the proper micrographs to the manuscript.

Reviewer 2 Report

The work reported in this manuscript (Influence of sodium hypochlorite treatment on pore size distribution of polysulfone/polyvinylpyrrolidone membranes) is interesting and well presented. However, it needs improvements before acceptance. The work requires a minor revision.

Comment 1: There are some typographical errors in the manuscript, so authors need to correct it in the revised manuscript.

Comment 2: The authors need to add and discuss some more literature review in the introduction section to strengthen the background of their work.

Comment 3: In SEM results: The authors should explore and discuss better their results with some more references in order to prepare a better discussion.

Author Response

Thank you very much for your time and effort to review our manuscript. We have made our best to improve our manuscript according to your comments. We have expanded Methods, Results and Discussions sections. We have also attended to the formatting and language of the manuscript according to your suggestions. Please note that your comments are in black font and our responses are in red font.

Comment 1. There are some typographical errors in the manuscript, so authors need to correct it in the revised manuscript.

Answer 1. We have attended to the formatting and language of the manuscript according to your suggestions.

Comment 2: The authors need to add and discuss some more literature review in the introduction section to strengthen the background of their work.

Answer 2. According to your suggestion we have improved the introduction and added new references:

  1. Li, K., Li, S., Su, Q., Wen, G., Huang, T. Effects of Hydrogen Peroxide and Sodium Hypochlorite Aging on Properties and Performance of Polyethersulfone Ultrafiltration Membrane. Int. J. Environ. Res. Public Health. 2019, 16(20), 3972. https://doi.org/10.3390/ijerph16203972.
  2. Hanafi, Y., Szymczyk, A., Rabiller-Baudry, M., Baddari, K. Degradation of poly (ether sulfone)/polyvinylpyrrolidone membranes by sodium hypochlorite: insight from advanced electrokinetic characterizations. Environ. Sci. Technol. 2014, 48(22), 13419-13426. https://doi.org/10.1021/es5027882.
  3. Causserand, C., Pellegrin, B., Rouch, J. C. Effects of sodium hypochlorite exposure mode on PES/PVP ultrafiltration membrane degradation. Water research 2015, 85, 316-326. https://doi.org/10.1016/j.watres.2015.08.028.
  4. Chokki, J., Darracq, G., Pölt, P., Baron, J., Gallard, H., Joyeux, M., Teychené, B. Investigation of Poly (ethersulfone)/Polyvinylpyrrolidone ultrafiltration membrane degradation by contact with sodium hypochlorite through FTIR mapping and two-dimensional correlation spectroscopy. Polym. Degrad. Stab. 2019, 161, 131-138. https://doi.org/10.1016/j.polymdegradstab.2019.01.017.
  5. Ravereau, J., Fabre, A., Brehant, A., Bonnard, R., Sollogoub, C., Verdu, J. Ageing of polyvinylidene fluoride hollow fiber membranes in sodium hypochlorite solutions. J. Membr. Sci. 2016, 505, 174-184. https://doi.org/10.1016/j.memsci.2015.12.063.
  6. Mavukkandy, M. O., Bilad, M. R., Giwa, A., Hasan, S. W., Arafat, H. A. Leaching of PVP from PVDF/PVP blend membranes: impacts on membrane structure and fouling in membrane bioreactors. J. Mater. Sci. 2016, 51(9), 4328-4341. https://doi.org/10.1007/s10853-016-9744-7.
  7. Qin, J. J., Li, Y., Lee, L. S., Lee, H. Cellulose acetate hollow fiber ultrafiltration membranes made from CA/PVP 360 K/NMP/water. J. Membr. Sci. 2003, 218(1-2), 173-183. https://doi.org/10.1016/S0376-7388(03)00170-4.

42. Qin, J. J., Cao, Y. M., Li, Y. Q., Li, Y., Oo, M. H., Lee, H. Hollow fiber ultrafiltration membranes made from blends of PAN and PVP. Sep. Pur. Technol. 2004, 36(2), 149-155. https://doi.org/10.1016/S1383-5866(03)00210-7.

Comment 3. In SEM results: The authors should explore and discuss better their results with some more references in order to prepare a better discussion.

Answer 3. We have added the discussion of SEM results with the previously reported data:

The results obtained in this work employing the SEM technique conform to the previously reported data. In the work [41], after the treatment of Cellulose acetate/PVP membrane with 200 ppm hypochlorite at pH=7 for over 24 h, the surface became smoother and the pore radius increased. After the exposure of the PSf/PVP membrane to a 4000 ppm hypochlorite solution at pH=7 for over 48 h, the inner surface became rough and the pore size on the outer surface increased [31]. In the work [35], employment of 5000 ppm NaOCl solution for 100 h at pH=9 led to a significant increase of the polyethersulfone/PVP membrane pore size. The authors of work [20] have noticed that after the treatment of PSf/PVP membranes with 100 ppm of HClO for 16 days at 25 °C the microscopic morphology changed from filamentous to spongy. In this work, the initial structure of the membrane was spongy and only treatment with 50 ppm led to the transformation of the inner surface from spongy to slightly filamentous due to the simultaneous removal of PSf and PVP. The roughness of inner and outer surfaces did not vary significantly with the increase of the pore size and porosity.

Round 2

Reviewer 1 Report

The authors have carefully revised their manuscript according to the comments of

the reviewers. Conequently, I recommend publication of the manuscript.